# Long COVID Clusters of Symptoms Persist beyond Two Years after Infection: Insights from the CARDIO COVID 20–21 Registry

**DOI:** 10.3390/v16071028

**Published:** 2024-06-26

**Authors:** Juan Pablo Arango-Ibanez, Brayan Daniel Córdoba-Melo, Juliana María Gutiérrez Posso, Mario Miguel Barbosa-Rengifo, Cesar J. Herrera, Miguel Angel Quintana Da Silva, Andrés Felipe Buitrago, María Lorena Coronel Gilio, Freddy Pow-Chong-Long, Juan Esteban Gómez-Mesa

**Affiliations:** 1Centro de Investigaciones Clínicas (CIC), Fundación Valle del Lili, Cali 760032, Colombia; juan.arango.ib@fvl.org.co (J.P.A.-I.); brayan.cordoba.me@fvl.org.co (B.D.C.-M.); mario.barbosa@fvl.org.co (M.M.B.-R.); 2Departamento de Cardiología, Centros de Diagnóstico y Medicina Avanzada y de Conferencias Médicas y Telemedicina (CEDIMAT), Santo Domingo F3QG+PJ6, Dominican Republic; 3Departamento de Cardiología, Instituto Cardiovascular Sanatorio MIGONE, Asunción 1541, Paraguay; miguelquintanadasilva@gmail.com; 4Departamento de Cardiología, Fundación Santa Fe, Bogota 110111, Colombia; 5Departamento de Cardiología, Instituto de Cardiología J. F. Cabral, Corrientes 3400, Argentina; 6Departamento de Cardiología, Hospital Luis Vernaza, Guayaquil 090306, Ecuador; dr.freddypowchl@gmail.com; 7Facultad de Ciencias de la Salud, Universidad Icesi, Cali 760031, Colombia; 8Departamento de Cardiología, Fundación Valle del Lili, Cali 760032, Colombia

**Keywords:** COVID-19, long COVID, post-COVID condition, cluster, symptoms, Latin America

## Abstract

Long COVID presents with diverse symptoms after COVID-19. Different clusters of symptoms have been reported; however, their persistence beyond 2 years after COVID-19 remains unclear. In this cohort study, we prospectively evaluated individuals with previous severe COVID-19 presenting with long COVID at a two-year follow-up. We characterized the included patients and performed a cluster analysis of symptoms through multiple correspondence analysis and hierarchical clustering. A total of 199 patients with long COVID were included. The median age was 58 years (48–69), 56% were male, and the median follow-up time since the COVID-19 diagnosis was 26 months (IQR: 25, 27). Three symptom clusters were identified: Cluster 1 is characterized by fatigue, myalgia/arthralgia, a low prevalence of symptoms, and a lack of specific symptoms; Cluster 2 is defined by a high prevalence of fatigue, myalgia/arthralgia, and cardiorespiratory symptoms, including palpitations, shortness of breath, cough, and chest pain; and Cluster 3 is demonstrated a high prevalence of ageusia, anosmia, fatigue, and cardiorespiratory symptoms. Our study reinforces the concept of symptom clustering in long COVID, providing evidence that these clusters may persist beyond two years after a COVID-19 diagnosis. This highlights the chronic and debilitating nature of long COVID and the importance of developing strategies to mitigate symptoms in these patients.

## 1. Introduction

Coronavirus Disease 2019 (COVID-19), caused by severe acute respiratory syndrome coronavirus 2 (SARS-CoV-2), first appeared in 2019. By December 2023, approximately 772 million cases and nearly 7 million deaths had been reported worldwide [1]. The long-term consequences following infection are collectively referred to as ‘long COVID’, also named ‘post-COVID-19 condition/syndrome’, and have garnered widespread attention due to its high incidence and difficult recognition [2]. Different clinical definitions have also been proposed for long COVID [3]. According to the World Health Organization (WHO), it is a condition that occurs in individuals with a history of probable or confirmed SARS-CoV-2 virus infection, three months after the onset of COVID-19, whose manifestations persist for at least two months, impact patients’ lives, and cannot be explained by an alternative diagnosis [4]. The Centers for Disease Control and Prevention (CDC) defines it as a spectrum of physical and psychological consequences in patients that occur four or more weeks after COVID-19 infection [5].

The signs and symptoms present in patients with long COVID are multisystemic and fluctuating, including fatigue, cognitive dysfunction, headache, post-traumatic stress, sleep disturbances, dyspnea, cough, myalgias, arthralgias, anosmia, dysgeusia, cardiac and gastrointestinal manifestations, among others [6,7]. This spectrum of manifestations is more commonly seen in patients with a history of severe COVID-19 and tends to occur in clusters, including neurological, cardiorespiratory, and systemic-inflammatory [8,9,10]. Furthermore, it includes not only multisystemic symptoms but also specific medical conditions like type 2 diabetes mellitus, encephalomyelitis, postural orthostatic tachycardia syndrome, and cardiovascular diseases, among others. The pathophysiology of long COVID includes a combination of factors like persistent SARS-CoV-2 reservoirs, immune dysregulation, and autoimmunity, leading to widespread organ damage and inflammation. It involves significant vascular and endothelial dysfunction, contributing to thrombosis and altered blood flow. Many of these changes may persist for several months [9].

Latin America (LATAM) has been greatly impacted by the COVID-19 pandemic, representing around 15% of the cases globally and 28% of the deaths by 2022 [11]. Moreover, research has shown that people with Hispanic or Latino Heritage present a higher prevalence of long COVID [12]. Currently, a gap exists in the information about long COVID in LATAM [13]. This has been highlighted by a recent narrative review [14] and a study suggesting that the impact of long COVID is underreported in this region [15]. Therefore, we decided to conduct a study involving patients from the Latin American Registry of Cardiovascular Disease and COVID-19 (CARDIO COVID 19–20) [16]. This study aims to investigate the characteristics and clusters of symptoms in individuals experiencing long COVID, persisting beyond two years after a severe COVID-19 diagnosis. Understanding the temporal extent and clustering of long COVID symptoms is crucial for developing comprehensive patient care and management strategies. Moreover, identifying different clusters of symptoms aids in understanding the different pathophysiological mechanisms that have been studied and will pave the path for specific therapy for each phenotypic presentation [9,10].

## 2. Materials and Methods

### 2.1. Study Design and Participants

We conducted a prospective cohort study involving patients who had COVID-19. These patients were identified from the CARDIO COVID 19–20 registry [16], a multicenter database comprising 3260 patients hospitalized with microbiologically confirmed COVID-19 from 44 institutions across 14 countries. Institutions participating in CARDIO COVID 19–20 were invited to join a follow-up study named CARDIO COVID 20–21. This follow-up study aimed to assess long-term symptoms, biomarkers, radiological abnormalities, and psychiatric disturbances, among other objectives. Both registries were coordinated and supervised by the Inter-American Council of Heart Failure and Pulmonary Hypertension (CIFACAH) of the Inter-American Society of Cardiology (IASC).

Inclusion criteria;Previous severe COVID-19 during hospitalization;Patients meeting the CDCs criteria for long COVID;Patients with symptoms either currently or within the last 3 months prior to the follow-up;Patients who signed informed consent to participate.Exclusion criteria included:Patients without symptoms.

### 2.2. Definitions

Severe COVID-19 was defined as;The need for intensive care unit (ICU) admission, or;Myocardial injury (evidenced by elevated troponin levels), or;High risk of venous thromboembolism (indicated by elevated D-dimer), or;Other de novo cardiovascular complications observed during the hospital stay, including acute heart failure, stroke, and pulmonary embolism.

This definition was used for the Cardio COVID 19–20 registry and was designed before the development of official international severity classifications [16]. Long COVID was defined as the presence of symptoms lasting more than four weeks following a COVID-19 diagnosis and not explained by other conditions. This definition aligns with that provided by the CDC [5].

### 2.3. Data Collection

For the CARDIO COVID 20–21 registry, all institutions that participated in the CARDIO COVID 19–20 (44 institutions from 14 countries) were invited to participate in this long-term follow-up. Six institutions from 5 countries agreed to participate.

Patients were initially contacted by phone and invited to participate. If the patient agreed to participate, informed consent was signed, and an in-person or virtual clinical evaluation was then performed.

Data on demographics, comorbidities, and other variables (e.g., COVID-19 vaccination status) were assessed at the follow-up. Physicians involved in the study conducted physical examinations, which included measuring vital signs and evaluating for signs of heart failure. We used a standardized virtual questionnaire to assess self-reported symptoms meeting the criteria for long COVID in participants; trained physicians of the research group supervised this evaluation. In this questionnaire, participants were asked about the presence of persistent symptoms following COVID-19 hospitalization, specifically those that remained until at least the last three months during the follow-up assessment. Thus, we aimed to evaluate the persistent symptoms of long COVID beyond two years of COVID-19 diagnosis. Symptoms evaluated in the questionnaire included: fatigue, myalgia/arthralgia, anorexia, palpitations, chest pain, angina, orthostatism, shortness of breath, cough, dysphagia, diarrhea, dry mouth, ageusia, and anosmia. These symptoms were selected due to their high prevalence in long COVID [10] and were defined as highly relevant by the participating researchers because of their practice experience.

### 2.4. Statistical Analysis

A descriptive analysis was conducted using the median and interquartile range (IQR) to summarize continuous variables, while frequency and percentage are described as categorical variables. The assumption of normality was assessed using the Wilk-Shapiro test. For the cluster analysis, our methodology was adapted from a study investigating patient clusters in long COVID [7].

First, a multiple correspondence analysis (MCA) was employed to identify symptom groups, reduce complexity, and eliminate redundancies, in this case, 14 symptoms. This method transforms categorical data into coordinates within a multidimensional space, using the χ2 distance between coordinates to group similarly close individuals. Various dimensional solutions are obtained from this technique. The choice of the optimal solution is based on the fewest dimensions needed to explain the maximum total variability, resulting in a reduction in the number of variables required to summarize the data. In this study, MCA was applied to symptoms present in at least 10% of the subjects, and less frequent symptoms were grouped into single variables.

Secondly, the MCA coordinates were subjected to agglomerative hierarchical clustering (HC) using the Ward minimum-variance method. Initially, this method considers each participant as an individual cluster and then gradually merges clusters based on the shortest Euclidean distance between them. This merging continues until only one cluster remains, containing all participants. To determine the most suitable number of clusters, the sum of within-cluster inertia was calculated for each partition. The partition exhibiting a substantial relative reduction in inertia was chosen as the optimal clustering point. This procedure was performed using the ‘FactoMineR’ R Studio package [17].

Fisher’s exact test and Kruskal–Wallis Rank Sum test were used to compare variables across clusters. We considered statistical significance to be a *p*-value less than 0.05. The statistical analysis was performed using R software version 2023.12.0 + 369. Figures were created using R software version 2023.12.0 + 369, Microsoft Excel version 2311, and Lucidchart.

## 3. Results

The CARDIO COVID 19–20 registry included 3260 patients. 869 died during hospitalization, 417 had moderate COVID-19, 37 died at the 1-month follow-up and 318 were unreachable at this 1-month follow-up (Figure 1).

For the current registry, CARDIO COVID 20–21, six institutions from five countries (Argentina, Colombia, Ecuador, Dominican Republic, and Paraguay) accepted to participate in this 2-year follow-up (Figure 2). Of the 1619 patients initially eligible, 1105 were excluded due to various barriers faced by their institutions. These included restricted access to stored clinical files, insufficient staffing for the project, and the temporary closure of some facilities owing to the health crisis, among other factors. Of the 514 remaining patients, 242 were excluded as they could not be reached or declined participation. 272 accepted to participate. For this subanalysis, 73 patients were excluded due to the absence of symptoms consistent with long COVID, resulting in 199 patients for the final analysis (Figure 1 and Figure 2).

Baseline clinical characteristics and comorbidities are shown in Table 1. The median age was 58 years (IQR: 48–69), and 111 (56%) were men. 114 (57%) patients were admitted to the ICU during acute COVID-19. The median time between COVID-19 diagnosis and follow-up was 26 months (IQR: 25, 27). The most frequent comorbidities included hypertension in 111 (56%) patients, overweight/obesity in 100 (51%) patients, and diabetes mellitus in 57 (29%) patients. 187 (94%) of patients reported a complete vaccination schedule for COVID-19 at the time of the follow-up, and this was considered complete when the patient had at least two doses of an mRNA vaccine (e.g., Pfizer-BioNTech or Moderna) or a single dose of a viral vector vaccine (e.g., Johnson & Johnson’s/Janssen). Patients included had no reinfection of COVID-19.

Table 2 shows the findings from the physical examinations. The median body mass index was 27.1 (IQR: 24.9, 30.1). Median vital signs were within normal range. Peripheral edema was observed in 22% of patients, and jugular regurgitation in 2.9%. Additionally, 15.5% of patients had a New York Heart Association (NYHA) functional classification scale of III or IV.

For the cluster analysis, four symptoms were combined: chest pain with angina and palpitations with orthostatism. These groups were subsequently labeled, respectively, as chest pain and palpitations. The five most frequent symptoms were fatigue (71%), myalgias/arthralgias (56%), shortness of breath (34%), palpitations (31%), and dry mouth (28%) (Table 3). Through MCA and HC performed on symptoms, three clusters were revealed (Figure 3).

In Cluster 1, there were 126 participants, and the predominant manifestations were fatigue (61%), myalgias/arthralgias (44%), and shortness of breath (24%). Patients in Cluster 1 had a lower prevalence of symptoms compared to patients in Clusters 2 and 3. Cluster 2 comprised 56 patients, and the most frequent symptoms included fatigue (91%), myalgias/arthralgias (84%), dry mouth (61%), palpitations (59%), cough (55%), and shortness of breath (55%). Cluster 3 included 17 patients, and the most frequent symptoms included ageusia (94%), anosmia (88%), fatigue (82%), palpitations (65%), and chest pain (65%).

We named Cluster 1 as “systemic-inflammatory”, Cluster 2 as “musculoskeletal-cardiorespiratory”, and Cluster 3 as “neurological-cardiorespiratory” according to the most representative symptoms present in each. There was an overall statistical significance for differences in all symptoms across clusters. The median number of symptoms was 2 (IQR: 1, 3) for Cluster 1, 5 (IQR: 4.75, 6.25) for Cluster 2, and 6 (IQR: 6, 9) for Cluster 3, showing a significant difference (*p* < 0.001).

Regarding patient characteristics by clusters, male sex showed a marked difference: only 3 (18%) patients in Cluster 3 were male, compared to 75 (60%) in Cluster 1 and 33 (59%) in Cluster 2 (*p* = 0.005). No other demographics or comorbidities displayed a statistically significant difference between clusters. However, in the case of findings at the physical exam (Table 2), a difference was seen in diastolic pressure, peripheral edema, and NYHA. The median diastolic pressure was 80 (IQR: 71, 84) in Cluster 1, 82 (IQR: 72, 87) in Cluster 2, and 70 (IQR: 60, 78) in Cluster 3 (*p* = 0.01). Peripheral edema was present in 10 (9.7%) patients from Cluster 1, 15 (28%) from Cluster 2, and 13 (75%) from Cluster 3 (*p* < 0.001). NYHA III or IV was present in 15 (12%) patients from Cluster 1, 9 (16%) from Cluster 2, and 6 (35%) from Cluster 3 (*p* < 0.03).

A subsequent analysis was performed to differentiate clusters 2 and 3, given the similar symptom profile (Table 4). A significant difference was detected in myalgia/arthralgia (*p* = 0.018), anorexia (*p* = 0.03), and especially in ageusia and anosmia (*p* < 0.001).

## 4. Discussion

This is a prospective, multicentric cohort study of Latin American patients with long COVID after an episode of severe COVID-19. We identified 199 patients with symptoms consistent with long COVID persisting for 2 years. Three clusters were observed through MCA and hierarchical clustering: Cluster 1 (systemic-inflammatory), Cluster 2 (musculoskeletal-cardiorespiratory), and Cluster 3 (neurological-cardiorespiratory). Evidence already exists for long COVID symptoms lasting over two years [18,19,20,21]; however, most studies evaluating clusters of symptoms have assessed them within a year or less [7,22,23,24,25,26,27,28,29,30,31,32,33,34]. To our knowledge, this is the first study in the literature demonstrating the persistence of long COVID clusters of clinical manifestations beyond two years after COVID-19 diagnosis. Other studies have reported similar findings, for instance, Ito et al. described clusters at a follow-up of 12 months [30], and Zhao et al. at a follow-up of 20 months [35]. The prolonged duration of symptoms could be related to the severity of acute illness, comorbidities, and age, among other factors [30,36,37,38]. These results highlight the substantial morbidity of long COVID. They also aid in recognizing symptom clusters, promoting physician awareness, and developing long-term management plans for affected patients.

In our study, the predominant symptoms observed were fatigue, myalgia/arthralgia, dry mouth, palpitations, shortness of breath, and cough. Notably, these symptoms overlap significantly with the findings from a meta-analysis including 76 studies, in which the most common symptoms that corresponded with ours were fatigue (37.8%), dyspnea (23.4%), muscle pain (10.2%), and anosmia (11.2%) [10]. This highlights a similarity between some of the most frequent symptoms reported in the literature and ours.

The markedly high prevalence of symptoms observed in our study suggests an increased risk of developing a long COVID phenotype with a higher number of symptoms in patients who had severe COVID-19. This observation aligns with findings from previous research that noted a similar trend and highlighted the relevance of monitoring for long COVID in patients with a history of severe disease [8,39]. Despite our assessment specifically targeting symptoms emerging after acute illness, the increased symptom prevalence observed in our study may also derive from a cohort with higher comorbidity rates, possibly resulting in symptom overlap between long COVID and pre-existing comorbidities. Moreover, this is a very specific group of patients, as we included only those with symptoms persisting beyond two years, potentially excluding those who had resolved long COVID.

Our clustering presents similar findings to those from a meta-analysis, which reported on eight studies using cluster analysis, of which seven identified clusters of symptoms. This meta-analysis demonstrated the presence of three phenotypes: systemic-inflammatory, cardiorespiratory, and neurological [10]. Our Cluster 1 presents similar characteristics to the systemic-inflammatory cluster reported in the meta-analysis, which includes symptoms such as muscle pain, weakness, sleep disorders, and gastrointestinal symptoms (diarrhea in this case). In this cluster, specific symptoms are absent, and the prevalence of symptoms is low, findings that closely align with the results of the mentioned meta-analysis. Our results strongly resemble those found in the studies by Frontera et al. [40], Kenny et al. [7], and Mateu et al. [41], in which a cluster with a low prevalence of symptoms and a lack of specific symptoms was reported.

Our Cluster 2 aligns with the cardiorespiratory cluster identified in the meta-analysis, with both exhibiting a high prevalence of symptoms such as shortness of breath, chest pain, and palpitations [10]. A recent study of 341 long COVID patients revealed three distinct symptom clusters. The cluster with a moderate number of patients predominantly exhibited systemic and cardiorespiratory symptoms, resembling our Cluster 2 [41]. Ageusia, and anosmia, were highly specific to Cluster 3, aligning with the neurological cluster in the meta-analysis and highlighting long COVID impact on the nervous system [10]. Additionally, studies investigating long COVID symptoms have identified clusters with ageusia and anosmia correlations [24,30,31,32]. Interestingly, most patients in Cluster 3 were women (82%); this reinforces a meta-analysis finding that women are less likely to recover from taste and smell dysfunction [42].

Notably, our Cluster 3 presents a high prevalence of cardiorespiratory symptoms, a finding not observed in the meta-analysis by Kuodi et al. [10]. The distribution of symptoms in Cluster 2 (musculoskeletal-cardiorespiratory) and Cluster 3 (neurological-cardiorespiratory) appears remarkably similar. The only statistically significant differences were found in anosmia, dysgeusia, and anorexia. While findings vary significantly among studies, there is a notable consistency in reports of symptom clusters with similar types, numbers, and characteristics in long COVID cases.

The meta-analysis identifies the presence of overlapping symptoms across different clusters, something that also occurs in our study. In the present paper, this happened especially to fatigue and myalgia/arthralgia. While our clusters show symptom similarities with those in the reviewed meta-analysis, a full comparison is limited due to the exclusion of several symptoms in our study, such as headaches, sleep disorders, and hair loss (among many others). Additionally, notable differences were observed between our study and the meta-analysis. Although there are similarities between our Cluster 1 and the systemic-inflammatory cluster in the meta-analysis, the symptom profile appears different [10]. These findings emphasize the heterogeneity and complexity of long COVID, underlining the need for continued research to characterize its phenotypes more precisely.

Different pathophysiological mechanisms explain the multis-system effects and symptoms of long COVID. These include viral reservoirs, persistent inflammation, autoimmunity, endothelial dysfunction, and microbiome dysbiosis [9,43]. Additionally, persistent organ damage following COVID-19 has been observed in multiple organs, including the heart, lungs, liver, kidney, and brain [9,44]. These abnormalities may help explain the various clusters of symptoms reported in long COVID, which could result from different combinations of pathophysiological mechanisms and affected organs. Hence, systemic symptoms like fatigue and musculoskeletal pain may arise from systemic mechanisms, such as delayed viral clearance, chronic inflammation, hypoxia, and microvascular injury [43,45]. Cardiorespiratory symptoms may result from lasting damage to the lungs and heart, since pulmonary and cardiac injuries are key aspects of COVID-19; this damage is due to inflammation, blood clots, endothelial dysfunction, autoimmunity, and ongoing viral presence [43,46]. Finally, neurological damage from COVID-19, such as direct viral harm, extensive neuroinflammation, and endothelial damage, may contribute to the neurological symptoms of long COVID [43]. Smell and taste abnormalities are explained by local inflammation and long-term alterations in neurological structures [47,48]. Further studies should explore the specific pathophysiological pathways and organ damage linked to each symptom cluster and their interactions.

Most patients in this cohort were men, contrary to previous findings from a meta-analysis that show that women are at increased risk of long COVID [49]. This finding could be related to the increased risk of severe COVID-19 among men [50], which was one of the criteria for selecting patients for this study. We observed that most patient characteristics, such as age or comorbidities, were not significantly different across clusters. A difference was noted in male sex, diastolic pressure, peripheral edema, and NYHA III-IV. In contrast to our findings, a study found that female sex and being a healthcare worker were linked to two clusters. This study also found that age and ethnicity were not associated with clusters [7]. Another study also found no differences between clusters regarding demographic characteristics (age, sex, race) and comorbidities (hypertension, diabetes, chronic obstructive pulmonary disease) [40]. Other studies have found mixed significance for patient characteristics across clusters [30,35]. Further addressing the risk factors associated with clusters may enhance our understanding of long COVID phenotypes and their identification.

## 5. Strengths and Limitations

Notably, this is one of the few studies from a Latin American population to report clusters of symptoms. The inclusion of patients with a history of severe disease ensures a focused examination of this subgroup. Another strength is the prospective inclusion of patients from different institutions and different countries. Most patients were evaluated in-person, compared to many other studies in the literature that collected data through virtual surveys. Conducting an in-person evaluation may lead to a more accurate symptom evaluation, as the physician administering the questionnaire can provide explanations directly to the patient. Moreover, in-person evaluation allowed us to obtain data on vital signs and physical examination.

The generalization of findings could be limited by the varying definitions of COVID-19 severity and long COVID; by the time the CARDIO COVID 19–20 and CARDIO COVID 20–21 registries were designed, but the official World Health Organization classification for COVID-19 severity was not released. Moreover, many institutions did not participate in this follow-up registry, and many patients were lost to follow-up. One reason for this situation is that some hospitals were provisional hospitals established during the pandemic. By the time of this study, some of these hospitals were no longer operational, and others did not have access to medical files from the pandemic period.

The study does not include other long COVID symptoms reported in the literature, such as sleep disturbances or brain fog, among many others. There was no longitudinal follow-up of the symptoms. Potential recall bias of the self-reported symptoms and the absence of a detailed time for each symptom are also important caveats. Caution is needed when interpreting our findings because of the characteristics of our cohort: previous severe COVID-19 at a time when no vaccines were available, and the presence of long COVID symptoms lasting more than two years. This situation is particularly important, as vaccination is associated with a reduced risk of long COVID effects [51].

The combination of myalgia/arthralgia may hinder comparison with other studies and may overestimate the presence of each by itself. The overlapping of symptoms across clusters highlights the limitations of clustering techniques and the complexity of long COVID symptoms. Finally, the relatively small number of patients in cluster 3 and, to a lesser extent, in cluster 2, diminishes the power of the findings when comparing these groups.

## 6. Conclusions

Long COVID has emerged as a significant and persistent public health challenge, affecting a substantial portion of patients recovering from COVID-19. Our study reinforces the concept of symptom clustering in long COVID, contributing evidence to their persistence beyond two years after infection. We identified three distinct symptom clusters named according to the prevalence of their symptoms: Systemic-Inflammatory, Musculoskeletal-Cardiorespiratory, and Neurological-Cardiorespiratory. Recognizing and categorizing these clusters is crucial, as it may inform potential targeted treatment strategies and enhance our ability to manage the multifaceted nature of long COVID effectively.

## Figures and Tables

**Figure 1 viruses-16-01028-f001:**
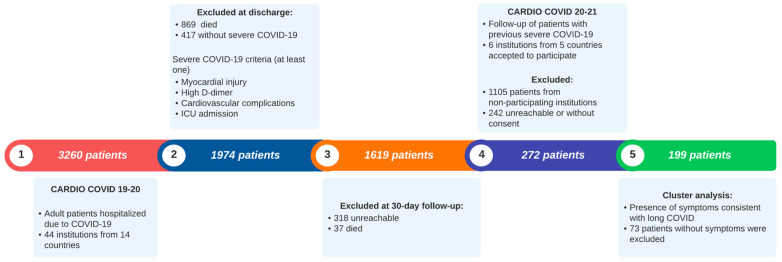
Flowchart of patient selection for CARDIO COVID 20–21 cluster analysis. Abbreviations: ICU, intensive care unit.

**Figure 2 viruses-16-01028-f002:**
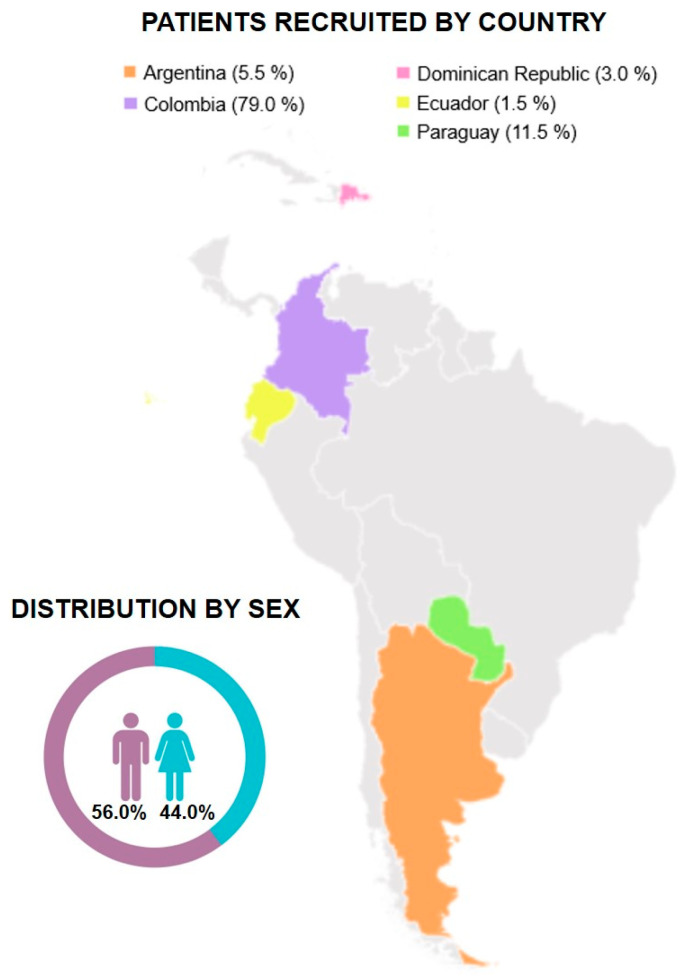
Patient recruitment by country and distribution by sex.

**Figure 3 viruses-16-01028-f003:**
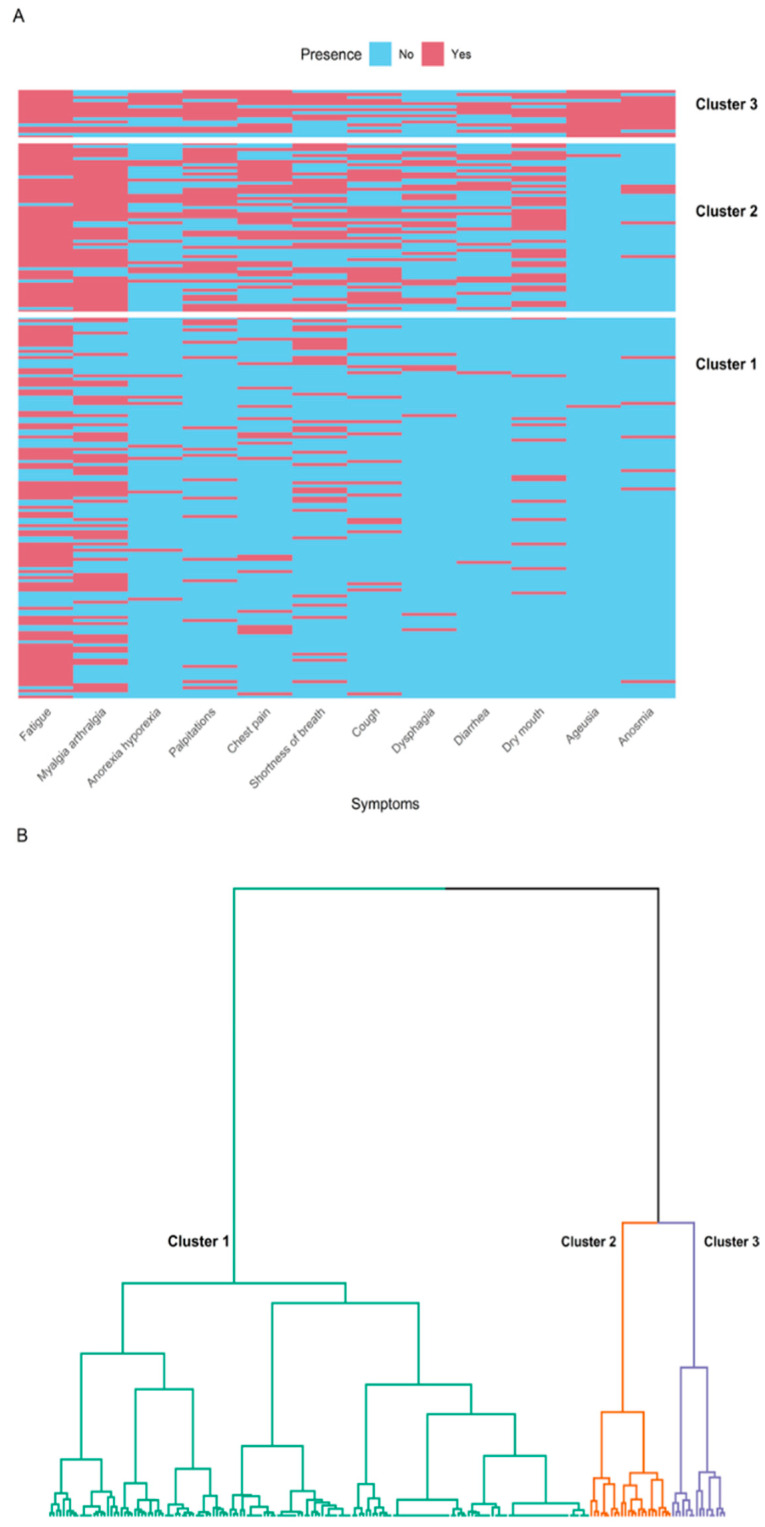
Dendrogram and heatmap of clusters of symptoms. (**A**): Heatmap showing the presence or absence of various symptoms in patients with long COVID, grouped into three distinct clusters (1, 2, and 3). Each row represents an individual patient, and each column represents a specific symptom. The presence of a symptom is indicated in red (Yes), while the absence is indicated in blue (No). (**B**): Dendrogram illustrating the hierarchical clustering of patients with long COVID based on their symptom profiles. The dendrogram shows three clusters: Cluster 1, Cluster 2, and Cluster 3.

**Table 1 viruses-16-01028-t001:** Demographics and comorbidities by cluster.

Characteristic	Overall	Cluster 1, n = 126	Cluster 2, n = 56	Cluster 3, n = 17	*p*-Value
Age (years)	58 (48, 69)	60 (47, 70)	56 (48, 64)	57 (53, 65)	0.6
Male	111 (56%)	75 (60%)	33 (59%)	3 (18%)	0.005
Full vaccine series	187 (94%)	116 (92%)	54 (96%)	17 (100%)	0.4
Admitted to the ICU	114 (57%)	71 (56%)	32 (57%)	11 (65%)	0.8
Outpatient follow-up after admission	157 (79%)	102 (81%)	41 (73%)	14 (82%)	0.5
Arterial hypertension	111 (56%)	68 (54%)	33 (59%)	10 (59%)	0.8
Overweight/Obesity	100 (51%)	65 (52%)	30 (55%)	5 (31%)	0.3
Diabetes mellitus	57 (29%)	29 (23%)	21 (38%)	7 (41%)	0.061
Dyslipidemia	23 (12%)	19 (15%)	4 (7.1%)	0 (0%)	0.10
Chronic kidney disease	26 (13%)	13 (10%)	9 (16%)	4 (26%)	0.089
Coronary artery disease	18 (9%)	14 (11%)	3 (5%)	1 (6%)	>0.9
Cancer (previous or active)	14 (7%)	7 (6%)	6 (11%)	1 (6%)	0.6
Asthma	14 (7%)	9 (7.1%)	3 (5.4%)	2 (12%)	0.5
Atrial fibrillation	13 (6.5%)	9 (7.1%)	3 (5.4%)	1 (5.9%)	>0.9
Heart failure	11 (5.5%)	7 (5.6%)	3 (5.4%)	1 (5.9%)	>0.9

Data are displayed as a number (%) or median (interquartile range). *p*-values were calculated using Fisher’s exact test, and Kruskal–Wallis Sum Rank test. Abbreviations: ICU, intensive care unit.

**Table 2 viruses-16-01028-t002:** Physical exam and clinical signs by cluster.

Variable	Number for Analysis	Total N = 199	Cluster 1, n = 126	Cluster 2, n = 56	Cluster 3, n = 17	*p*-Value
Body mass index (kg/m^2^)	186	27.1 (24.6, 30.5)	27.1 (24.9, 30.1)	28.2 (24.5, 32)	26 (25.2, 27.9)	0.7
Heart rate (pulse rate)	186	72 (66, 80)	72 (67, 80)	72 (67, 80)	68 (64, 72)	0.3
Systolic pressure (mmHg)	186	127 (115, 139)	128 (117, 139)	127 (115, 139)	116 (107, 148)	0.4
Diastolic pressure (mmHg)	186	80 (70, 85)	80 (71, 84)	82 (72, 87)	70 (60, 78)	0.01
Respiratory rate (bpm)	186	18 (17, 18)	18 (17, 18)	17 (17, 18)	17 (17, 18)	0.3
Oxygen saturation (%)	185	97 (96, 98)	96 (96, 98)	97 (95.5, 97)	97 (96, 97)	>0.9
Peripheral edema	174	38 (22%)	10 (9.7%)	15 (28%)	13 (76%)	<0.001
Jugular regurgitation	175	5 (2.9%)	4 (3.8%)	1 (1.9%)	0 (0%)	0.8
NYHA III-IV	199	30 (15.5%)	15 (12%)	9 (16%)	6 (35%)	0.03

Data are displayed as mean and standard deviation, median (interquartile range), or number (%). *p*-values were calculated using Fisher’s Exact Test and Kruskal–Wallis rank sum test. Abbreviations: NYHA, New York Heart Association Functional classification; bpm, breaths per minute. As some interviews were conducted virtually, the number for analysis is discordant with the total number of patients for some variables.

**Table 3 viruses-16-01028-t003:** Distribution of symptoms by Cluster.

Symptom	Total N = 199	Cluster 1, n = 126	Cluster 2, n = 56	Cluster 3, n = 17	*p*-Value
Fatigue	142 (71%)	77 (61%)	51 (91%)	14 (82%)	<0.001
Myalgia/arthralgia	112 (56%)	56 (44%)	47 (84%)	9 (53%)	<0.001
Anorexia	31 (16%)	8 (6.3%)	14 (25%)	9 (53%)	<0.001
Palpitations	61 (31%)	17 (13%)	33 (59%)	11 (65%)	<0.001
Chest pain	59 (30%)	18 (14%)	30 (54%)	11 (65%)	<0.001
Shortness of breath	68 (34%)	30 (24%)	31 (55%)	7 (41%)	0.002
Cough	54 (27%)	16 (13%)	31 (55%)	7 (41%)	<0.001
Dysphagia	34 (17%)	6 (4.8%)	24 (43%)	4 (24%)	<0.001
Diarrhea	26 (13%)	2 (1.6%)	17 (30%)	7 (41%)	<0.001
Dry mouth	56 (28%)	13 (10%)	34 (61%)	9 (53%)	<0.001
Ageusia	18 (9.0%)	1 (0.8%)	1 (1.8%)	16 (94%)	<0.001
Anosmia	26 (13%)	6 (4.8%)	5 (8.9%)	15 (88%)	<0.001

Data are displayed as a number (%). *p*-value was calculated with Fisher’s Exact Test for Count Data with simulated *p*-value (based on 2000 replicates); Kruskal–Wallis rank sum test.

**Table 4 viruses-16-01028-t004:** Differences in symptom distribution between Cluster 2 and 3.

Symptom	Cluster 2, n = 56	Cluster 3, n = 17	*p*-Value
Fatigue	51 (91%)	14 (82%)	0.4
Myalgia/Arthralgia	47 (84%)	9 (53%)	0.018
Anorexia	14 (25%)	9 (53%)	0.03
Palpitations	33 (59%)	11 (65%)	0.7
Chest Pain	30 (54%)	11 (65%)	0.4
Shortness of breath	31 (55%)	7 (41%)	0.3
Cough	31 (55%)	7 (41%)	0.3
Dysphagia	24 (43%)	4 (24%)	0.2
Diarrhea	17 (30%)	7 (41%)	0.4
Dry mouth	34 (61%)	9 (53%)	0.6
Ageusia	1 (1.8%)	16 (94%)	<0.001
Anosmia	5 (8.9%)	15 (88%)	<0.001

Data are displayed as a number (%). *p*-value was calculated with Fisher’s Exact Test for Count Data with simulated *p*-value (based on 2000 replicates).

## Data Availability

The original contributions presented in the study are included in the article; further inquiries can be directed to the corresponding author.

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
