# Peer review of "Long COVID Clusters of Symptoms Persist beyond Two Years after Infection: Insights from the CARDIO COVID 20–21 Registry"

_viruses, 2024, doi:10.3390/v16071028_

Round 1
Reviewer 1 Report
Comments and Suggestions for Authors
this is an interesting paper on long COVID and clustering. The paper is well written. I think that the paper and the discussion would benefit of the inclusion of some hypotheses, mostly on underlying mechanisms if possible, about the explanations of each cluster. This would help to better understand why these clusters have emerged and potential therapeutic targets in the future
Author Response
" I think that the paper and the discussion would benefit of the inclusion of some hypotheses, mostly on underlying mechanisms if possible, about the explanations of each cluster. This would help to better understand why these clusters have emerged and potential therapeutic targets in the future"
- We truly appreciate your interest in our study. We agree that hypothesizing the underlying mechanisms behind clusters of symptoms in long COVID could enhance our manuscript. Consequently, we have added a paragraph discussing this topic.
Thank you once again for your review.
Reviewer 2 Report
Comments and Suggestions for Authors
This is a prospective study of Latin American patients to catalog symptoms that persist two years after an acute COVID infection. Three clusters of patients are identified that exhibit different cross-sections of the symptoms examined. The results are largely consistent with prior meta-analyses performed by others. However, this study is unique in studying Latin American patients specifically.
Strengths
- - This study was conducted prospectively
- - Patients were examined in person by a qualified physician
- - The original COVID infection was confirmed microbiologically
Weaknesses
- - Restricted to the most severely affected patients requiring ICU admission or
exhibiting cardiovascular symptoms during acute infection
- - All acute infections occurred prior to availability of vaccinations so the
effects of vaccination not accounted for
- - Only 199 patients
Other
- - Check that the correct number of participating institutions is cited: is it six
(as on page 4) or 7 (as on page 3)?
Author Response
"Check that the correct number of participating institutions is cited: is it six (as on page 4) or 7 (as on page 3)?"
- We appreciate your time and commitment in reviewing our manuscript. The number inconsistency you pointed out was indeed a mistake. We have corrected it to reflect the accurate information: six institutions from five countries.
Thank you once again for your review.